# The Effects of Vaccines on the Sequelae Rates of Recurrent Infections and the Severity of Pulmonary COVID-19 Infection by Imaging

**DOI:** 10.3390/vaccines11081321

**Published:** 2023-08-04

**Authors:** Suzan Bahadir, Ebru Kabacaoglu, Kemal Bugra Memis, Hasan Ilksen Hasan, Sonay Aydin

**Affiliations:** 1Department of Radiology, Fiona Stanley Hospital, 11 Robin Warren Drive, Murdoch, WA 6150, Australia; suzan.bahadir@health.wa.gov.au (S.B.); hasan.hasan@health.wa.gov.au (H.I.H.); 2Department of Chest Diseases, Baskent University Alanya Research and Training Center, Yunus Emre Avenue, Alanya 07400, Turkey; ekabacaoglu@baskent.edu.tr; 3Department of Radiology, Erzincan Binali Yildirim University, Basbaglar, 1429th Street, Erzincan 24100, Turkey; kemalbugra.memis@saglik.gov.tr

**Keywords:** coronavirus disease, thorax CT, vaccination, reinfection, lung complications, sequelae

## Abstract

Although vaccines have been shown to reduce the number of COVID-19 infection cases significantly, vaccine-related reactions, long COVID-19 syndrome, and COVID-19 infection following vaccination continue to be a burden on healthcare services and warrant further scientific research. The purpose of this study was to research the severity of pulmonary COVID-19 infection following vaccination and the sequelae rates of recurrent infections in vaccinated cases by imaging. Patients who underwent follow-up CTs at 1 month, 3 months, and 6 months in our hospital with a diagnosis of COVID-19 were scanned retrospectively. Furthermore, all essential information was gathered from patients’ immunization records. The major findings of our study were: (1) sequelae were frequently observed in unvaccinated cases; (2) the correlation between vaccination status and the severity of sequelae was significant; (3) there was not any significant relationship between the vaccine type and the severity of sequelae; and (4) hematocrit, hemoglobin, and lymphocyte parameters may be used as predictors of sequelae rates. COVID-19 infection, although reduced in prevalence following the development of vaccines, still remains a public health concern because of reinfection. Vaccination not only appears to protect against primary infection, but also seems to reduce reinfection and sequalae rates following reinfection.

## 1. Introduction

Coronavirus disease (COVID-19) is caused by severe acute respiratory syndrome coronavirus 2 (SARS-CoV-2). It was declared a pandemic in March 2020 and has affected millions of people since then. According to the World Health Organization (WHO), there have been more than 760 million confirmed COVID-19 cases, nearly 7 million related deaths, and a total of 13 billion vaccine doses administered [1]. This pandemic has clearly caused significant morbidity and mortality as well as a large burden on healthcare expenditure. Studies have shown that the first infection may cause pulmonary–extrapulmonary sequelae-related pathology and death [2,3,4,5,6,7,8].

Because previous SARS and MERS outbreaks indicated that symptoms and imaging abnormalities persist over time, it has been proposed to monitor patients following acute SARS-CoV-2 pneumonia. In 2020, the British Thoracic Society (BTS) published a document for post-COVID-19 care that distinguished between patients with severe pneumonia and those with mild–moderate pneumonia. The goal of this document was to standardize radiological follow-up and then reduce the pressures on respiratory services following the initial COVID-19 outbreak [9,10,11]. Further clarification is therefore required to direct further global healthcare initiatives and improve outcomes.

Previous studies have investigated lung sequelae of COVID-19, demonstrated that more than half of the recovered patients showed thorax CT abnormalities months after infection. Ground glass opacities (GGOs), parenchymal bands/fibrous stripes, and reticulation were particularly common [12].

Although there are many studies indicating that vaccination against COVID-19 decreases the severity of infection, hospitalization rates, and admission to intensive care units, re-infection still occurs. Vaccination also appears to have protective effects on children and adolescents [13].

Although there are a few studies investigating the vaccine’s effect on long COVID-19 syndrome and acute and post-acute sequelae, to our knowledge, published data are lacking regarding the long-term effects of vaccines on lung-sequelae-related pathology [10,14,15]. To date, no studies have approached this topic by evaluating imaging appearances. The current literature appears to be primarily based on clinical and laboratory outcomes. Our aim is to evaluate the effects of vaccines on lung sequelae and the severity of infection in the lungs through a single-center, observational study. We hope to fill the knowledge gap on this particular topic so that this knowledge might be utilized to decide whether vaccination should be maintained or not.

## 2. Materials and Methods

This study was carried out in a university hospital, which is a tertiary referral center in our country (Baskent University Hospital, Ankara, Turkey). We decided to study this topic in August 2021. We investigated 900 lung CTs of patients from the hospital’s PACS between April 2020 and August 2021 who were diagnosed with COVID-19 pneumonia. Patients who had follow-up CTs at 1 month, 3 months, and 6 months were included retrospectively. Among these 900 patients, 100 patients with presentation, 1st, 3rd, and 6th month follow-up thoracic CT examinations within the specified date ranges were included in our study. Patients admitted with COVID-19 pneumonia between August 2021 and November 2022 were followed up using the same protocol (i.e., a CT at presentation, 1 month, 3 months, and 6 months follow-up). A polymerase chain reaction (PCR) test was positive in all COVID-19 cases included in this study. The inclusion criteria were: diagnosis of COVID-19 pneumonia confirmed with PCR, having the abovementioned follow-up CTs, and hospital admission. For the vaccinated group, at least a single dose of vaccination was required. The control group consisted of unvaccinated cases satisfying the inclusion criteria. Patients with typical symptoms but negative PCR test results, patients with missing clinical, imaging, and laboratory data, and patients who took a voluntary hospital discharge were excluded from this study.

Data were collected regarding age, sex, contact history, coexistent diseases (hypertension, diabetes, chronic obstructive pulmonary disease, malignant disease, dementia, anemia, and other diseases), medications used, systolic and diastolic blood pressures, body temperature, and pulse rate. The presence or absence of arrhythmia and fingertip oxygen saturation were also recorded. Vaccination status, number of doses, implement dates, and the manufacturer of the vaccine were recorded. There were no nonoverlapping populations, including those aged 65 years or older, those with high-risk comorbid conditions, and those with immunocompromising conditions, in our study population.

Sinovac and Biontech vaccines are made according to the preferences of the people in our country. There is no indication determined by our Ministry of Health in this regard. The first vaccine supplied to our country during the pandemic period was Sinovac, which was supplied about 6 months before Biontech. For this reason, there are people who take a booster dose with Biontech after vaccination with Sinovac.

Baseline (within 0–3 days after the first CT) hemoglobin (Hb), hematocrit (Htc), white blood cell count (WBC), platelet count (plt), lymphocyte, neutrophil, neutrophil to lymphocyte ratio (NLR), C-reactive protein (CRP), D-dimer, ferritin, procalcitonin, and fibrinogen levels were obtained from the hospital’s laboratory records. Written informed consent was obtained from all patients regarding the use of their data in this scientific study. The Baskent University Ethics Committee application was used for this study. Baskent University’s approval number is: E-946603339-604.01.02-4007 (date: 19 January 2021).

All of the chest CT images were evaluated by 2 board-certified radiologists in consensus. Therefore, there was no inter-observer disagreement. A multi-detector CT scanner (Somatom Sensation 64; Siemens) was used for all examinations, and scanning parameters were standard, which are recommended by the pre-setting for thorax routine imaging. Diagnosis of COVID-19 pneumonia was considered when ground glass opacity, crazy-paving pattern, and consolidation were observed on the chest CT based on the standard lexicon for thoracic imaging reported by the Fleischner Society [16]. Sequelae were described as band-like atelectasis, traction bronchiectasis, and focal ground glass areas similar to other patterns of fibrosis mostly seen in peripheral/subpleural and basal parts of the lungs [17].

Based on the Guideline for the Diagnosis and Treatment Plan of COVID-19 Infection by the National Health Commission, pulmonary COVID-19 infection was classified into three types: mild to moderate type common with fever, respiratory symptoms, and imaging presentations of pneumonia; severe type with any of the following: respiratory distress with RR > 30 times/minutes, oxygen saturation at rest <93%, or PaO_2_/FiO_2_ < 300 mmHg (1 mmHg = 0.133 kPa); and critically severe type with any of the following: respiratory failure needing mechanical ventilation, shock, or in combination with other organ failure needing ICU intensive care.

### Statistical Aproach

Data were analyzed using SPSS for Windows, version 24.0 (SPSS Inc., Chicago, IL, USA). The Shapiro–Wilk test was used to test the distributions of continuous variables for normality. Descriptive statistics for continuous data are shown as the median and Q1 (percentile 25) and Q3 (percentile 75). Categorical data are shown as numbers and percentiles. The differences between groups were compared using Student’s t-test for means and the Mann–Whitney U-test for medians. Categorical data were analyzed using Pearson’s chi-square or Fisher’s exact test, as appropriate. A *p*-value of less than 0.05 was considered statistically significant. The cut-off values for parameters for discrimination between the sequelae groups were determined using ROC analysis (Htc, Hb, and lymphocyte count). For each value (Htc, Hb, and lymphocyte), the sensitivity and specificity of each outcome were studied.

## 3. Results

### 3.1. General Characteristics of the Cases

In this study, a total of 100 patients, of whom 43 were female and 57 were male, were included. The average age was 56 (range 44–65). Twenty-three patients with severe COVID-19 pneumonia (respiratory rate >30 breaths/min, SpO_2_ < 90% on room air or respiratory distress) were discharged from the hospital after treatment, and all of them were unvaccinated. There were 77 patients with mild to moderate COVID-19 pneumonia, 25 of whom were hospitalized. There were no deaths, respiratory failures, or patients who needed mechanical assistance. There were no sequelae in 74 patients. An obvious interstitial pattern was observed in 13 patients, and mild fibrotic changes were observed in the remaining 13 patients. The general socio-demographic and clinical characteristics of the cases are presented in Table 1.

### 3.2. Clinical Characteristics of Studied Groups

Individuals in the two groups had similar characteristics in terms of age and gender distribution (*p* > 0.05). The average lymphocyte count in cases with sequelae was 1.44 (range 1.11–1.90), which was significantly higher (*p* = 0.001) than the lymphocyte count in cases without sequelae, 1.12 (range 0.76–1.39).

A significant correlation was found between the status of being vaccinated and the occurrence of sequelae. The frequency of vaccination in cases without sequelae was found to be significantly higher, with 40 cases (54%), compared to the frequency of vaccination in cases with sequelae, with 4 cases (15.3%) (*p* < 0.001). Sequelae were frequently observed in unvaccinated cases.

A significant correlation was found between Sinovac vaccination and the sequelae rate (*p* = 0.001). Sequelae occurred in 96.7% of individuals who did not receive a vaccine. Moreover, sequelae did not occur in 41.4% of individuals who received a single dose or multiple doses of the vaccine.

The demographic and clinical characteristics of cases with and without sequelae are compared in Table 2.

### 3.3. Relationship between Sequelae Levels and Vaccination

A significant correlation was found between the vaccination status and the severity of sequelae (*p* = 0.005). When the frequency of vaccination (47.3%) in individuals without sequelae was compared with the frequency of vaccination in mild/moderate and severe sequelae cases (15.4% and 7.7%, respectively), statistically significantly higher rates of sequelae were observed in those who were not vaccinated. There was no significant relationship between vaccine type and severity of sequelae (*p* > 0.05). The relationship between the severity and rates of sequelae and vaccines is presented in Table 3.

Thoracic CT images of a 51-year-old unvaccinated patient are shown in Figure 1.

Thoracic CT images of a 45-year-old unvaccinated patient are shown in Figure 2.

Thoracic CT images of a 43-year-old patient vaccinated with a single dose of Biontech are shown in Figure 3.

Thoracic CT images of a 43-year-old patient vaccinated with three doses of Biontech are shown in Figure 4.

Thoracic CT images of a 54-year-old patient vaccinated with two doses of Sinovac are shown in Figure 5.

### 3.4. Relationship between Sequelae Status and Blood Parameters

The relationship between the presence of sequelae and blood parameters was examined by receiver operating characteristic (ROC) analysis (Table 4, Figure 6). When sequelae rates were correlated with Htc, Hb, and lymphocyte values, it was observed that the Area Under the Curve value was statistically significant (*p* < 0.05) for a positive correlation between higher values and more severe sequelae rates. Thus, it is thought that these three parameters may be used as markers for the prediction of sequelae in affected individuals.

When the Htc value was above 44.15, the sensitivity was 0.457, and the specificity was 0.806 for cases diagnosed with sequelae. Additionally, when the Hb value was above 13.95, the sensitivity was 0.536, and the specificity was 0.672. Finally, when the lymphocyte count was above 1.36, the sensitivity was 0.714, and the specificity was 0.746 for the prediction of sequelae.

## 4. Discussion

In this study, we compared lung changes following COVID-19 infection in terms of severity and sequelae rates confirmed through CT imaging in vaccinated and unvaccinated cases. Although rates of post-vaccine infections have decreased recently, optimal management remains unclear. Our study aims to provide further clarification with regard to this continually evolving clinical issue.

The major findings of our study were as follows: (1) sequelae were frequently observed in unvaccinated cases; (2) the correlation between vaccination status and the severity of sequelae was significant; (3) there was not any significant relationship between the vaccine type and the severity of sequelae; and (4) Htc, Hb, and lymphocyte parameters may be used as predictors of sequelae rates.

It is currently estimated that approximately one billion people have been infected with the coronavirus worldwide. Following the introduction of vaccines, case numbers have reduced significantly. However, concerns over reinfection rates following vaccines, and whether recurrent infection in vaccinated cases carries similar adverse health outcomes compared to unvaccinated cases, remain unclear. In one study, Bowe et al. demonstrated a cumulative risk with regard to increased mortality and adverse health outcomes related to repeated infection. In addition, a direct correlation has been demonstrated between the numbers of repeated reinfections and adverse outcomes compared to single episodes of the infection. Reinfection appears to confer an increased risk of adverse outcomes in both the acute and subacute phases.

Given the apparent ability of this virus to rapidly mutate, it is likely that this disease is going to be an ongoing global healthcare concern for the foreseeable future. The question of how many repeated vaccine doses are required to produce optimal protection remains unclear and is a matter of ongoing debate. This study does not address this issue, which should be investigated further.

In the time period when we planned our work, there was no study in the literature describing sequelae findings in imaging after COVID-19 pneumonia. In our study, sequelae were described as band-like atelectasis, traction bronchiectasis, and focal ground glass areas similar to other patterns of fibrosis mostly seen in peripheral/sub-pleural and basal parts of the lungs. These findings are similar to the most common sequelae of pulmonary parenchymal changes that Watanabe et al. described in their study [12]. In this regard, our study complies with standardization in the literature.

In our study, we found that following COVID-19 pneumonia, the frequency of vaccination in cases without sequelae was significantly higher (50%) compared to the frequency of vaccination with sequelae. Therefore, vaccination not only appears to protect against primary infection but also seems to decrease sequalae rates following reinfection. Several studies exist that appear to confirm this impression [18,19,20,21].

Our study also demonstrates a significant correlation between vaccination status and the severity of consequent sequelae. Post-infectious sequelae may affect many organ systems and have serious repercussions by creating a significant burden for healthcare systems [22]. Currently, there is insufficient evidence to explain the pathological mechanisms that lead to pulmonary fibrosis and subsequent sequalae. For instance, in one study, Kumar et al. suggested that aberrant myofibroblast proliferation may be responsible for fibrotic lung injury with accompanying perivascular inflammation and thromboembolic phenomenon [23]. As the lungs are the primary focus of infection, it is not surprising that pulmonary-fibrosis-related respiratory symptoms and death are frequently observed [24]. A number of studies exist that suggest that vaccination may have a protective effect against subsequent lung fibrosis and its complications, but the pathological mechanism of this effect is unsubstantiated [10,13,18,19,20,21,22].

For example, Gao et al. have confirmed in a systematic review of 18 separate studies that two doses of the vaccine appear to be protective against long COVID-19, while one dose does not appear to be so [10]. A prospective study by the Dutch group, led by Wyenberg et al., showed no evidence of beneficial effects of vaccination on long COVID-19 symptoms, which was corroborated by serological data based on neutralizing antibodies [14]. We suggest that these findings may be related to only a single dose of the vaccine being administered to the study population. It appears, therefore, that at least two courses of vaccine are required to confer a beneficial effect. In our study, we did not observe any significant difference between the vaccine type (Sinovac and Biontech in our study) and the severity of sequelae.

We found a mortality rate of 1.1% in the 900 patients we screened. According to the WHO dashboard, although this rate is above the rate stated for Turkey, it is similar to the global mortality rates.

Finally, it appears that Htc, Hb, and lymphocyte count may be useful markers for predicting adverse sequelae following COVID-19 infection.

This study has some strengths: to our knowledge, this is one of the first studies to evaluate the severity of pulmonary infection and sequalae rates by CT imaging. Moreover, prolonged imaging follow-up, blood results combined with imaging findings, and clinical data provided a more robust assessment. Our study has some limitations as well. Firstly, it is a single-center study with a limited number of patients. Although our results are statistically powerful, larger cohort studies are required from multiple centers to further confirm our findings. Another issue is that at the time of this study, all the variants of COVID-19 were not subtyped well; thus, new study sets should be applied for each variant to provide a better understanding of variant behaviour. So, our study does not address in detail symptomatic COVID-19 pneumonia due to possible variant virus infection confirmed by PCR after vaccination boosters. Another issue is that it is an undeniable fact that 6-month short-term imaging results will not be sufficient for evaluation in the follow-up of pulmonary sequelae after pulmonary COVID-19 infection. However, we did not have sufficient data to evaluate the long-term results, as most of the patients stopped coming to the hospital after the end of the pandemic. In addition, we generally conducted this study on patients admitted to the hospital with thoracic CT imaging, although, rarely, home-treated patients came to the hospital for thoracic CT control and were included in our study. However, almost all mild or asymptomatic COVID-19 patients were excluded from the sample because they were treated at home. In addition, there is a systematic sampling bias because patients with severe and critical COVID-19 pneumonia, most of which resulted in death and disability, could not be included in our study due to the lack of 6-month follow-up images. The inclusion criterion was not a standard protocol. The sample was taken from patients who were admitted to our hospital. These findings may not be generalizable because they only comprise patients who had follow-up thoracic CT scans. Lastly, as the examined blood parameters were acquired within three days of the first diagnosis, we cannot offer any results about the temporal change of these parameters and/or the relationship between this change and the sequelae formation.

Future research should focus on vaccine prevention against variant subtypes, sequalae rates, and the pathogenesis of sequalae in all organ systems. When pathogenetic pathways are identified, targeted treatment options could be developed to achieve sustainable protection.

## 5. Conclusions

In conclusion, our study showed that a high frequency of residual CT abnormalities and sequelae findings are seen more frequently in unvaccinated cases months after COVID-19. Even though we cannot evaluate the combined effect of multiple laboratory parameters on sequelae formation, with the presented data, one can infer that vaccination can reduce the rate of sequelae following possible infection. Further investigations with longer follow-up periods will help in understanding the underlying mechanism of COVID-19 sequelae and in managing patients with these conditions.

## Figures and Tables

**Figure 1 vaccines-11-01321-f001:**
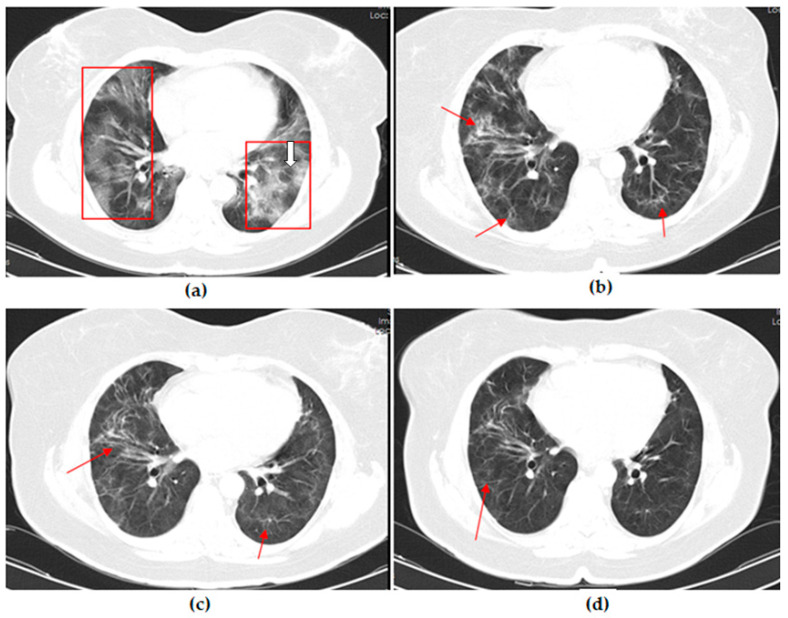
A 51-year-old unvaccinated female COVID-19 patient presenting with fever with dry cough for 4 days. (**a**) The axial thorax CT image shows a diffuse ground glass opacity (GGO) appearance in the bilateral lower lobes and right middle lobe (red frames) and a reversed halo sign (white arrow) in the lateral basal segment of the left lower lobe at presentation. (**b**) The 1st month, (**c**) 3rd month, and (**d**) 6th month follow-up axial thorax CT images show that GGO appearance of the bilateral lower lobes is lesser compared to the previous CT scan. Multiple fibrotic parenchymal bands (red arrows) were found, especially in the anterior basal segment of the right lower lobe.

**Figure 2 vaccines-11-01321-f002:**
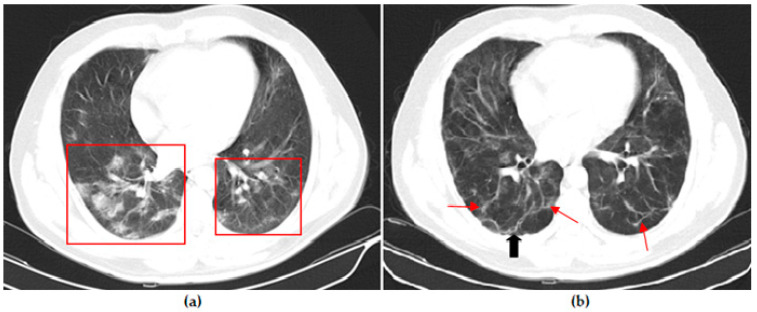
A 45-year-old unvaccinated male COVID-19 patient presenting with fever with diarrhea for 3 days. (**a**) The axial thorax CT image shows a patchy ground glass opacity (GGO) appearance in the bilateral lower lobes (red frames) at presentation. (**b**) The 1st month follow-up axial thorax CT image shows that the GGO appearance of the bilateral lower lobes is lesser than the previous CT scan. Multiple fibrotic parenchymal bands (red arrows) and mild dilatation of the distal bronchial branches (black arrow) were found, especially in the superior segment of the right lower lobe.

**Figure 3 vaccines-11-01321-f003:**
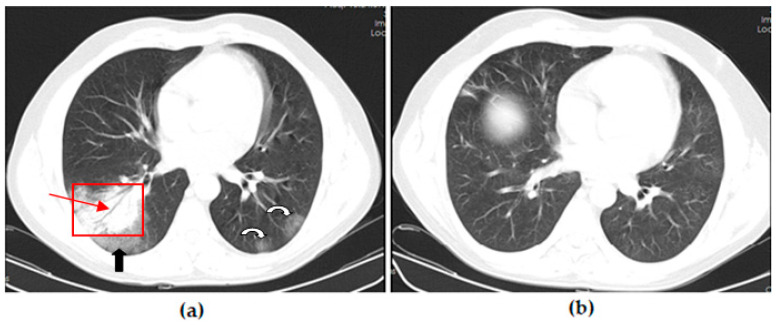
A 43-year-old male COVID-19 patient vaccinated with a single dose of Biontech presenting with fever with myalgia for 4 days. (**a**) The axial thorax CT image shows ground glass opacities (black arrow) around consolidation areas (red frame), including an air bronchogram (red arrow) in the lateral basal segment of the right lower lobe and a patchy GGO appearance in the posterior basal segment of the left lower lobe (white curved arrow) at presentation. (**b**) The 1st month follow-up axial thorax CT image shows consolidation areas, the GGO appearance regressed, and no pathological findings were found.

**Figure 4 vaccines-11-01321-f004:**
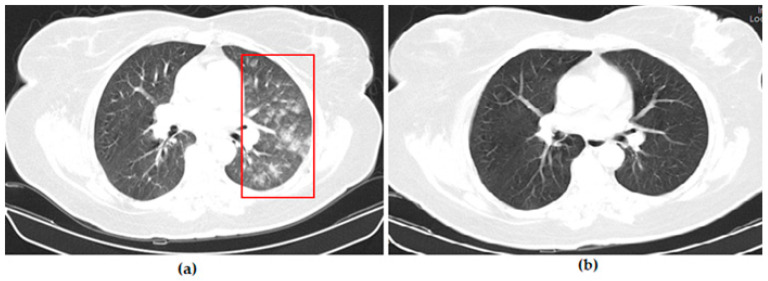
A 53-year-old female COVID-19 patient vaccinated with three doses of Biontech presenting with fever for 3 days. (**a**) The axial thorax CT image shows a patchy ground glass opacities appearance (red frame) in the left upper lobe at presentation. (**b**) The 1st month follow-up axial thorax CT image shows the GGO appearance regressed, and no pathological findings were found.

**Figure 5 vaccines-11-01321-f005:**
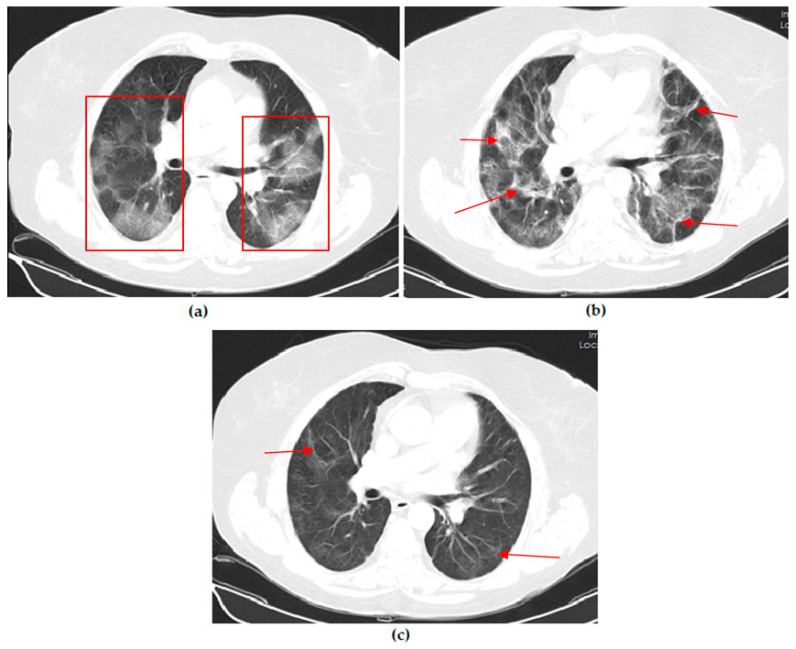
A 54-year-old female COVID-19 patient vaccinated with two doses of Sinovac presenting with fever for 7 days. (**a**) The axial thorax CT image shows a patchy ground glass opacity (GGO) appearance in the bilateral upper lobes and the superior segment of both lower lobes (red frames) at presentation. (**b**) The 1st month and (**c**) 6th month follow-up axial thorax CT images show the GGO appearance of the bilateral lower lobes is lesser than previous CT scan. Multiple fibrotic parenchymal bands (red arrows) were found.

**Figure 6 vaccines-11-01321-f006:**
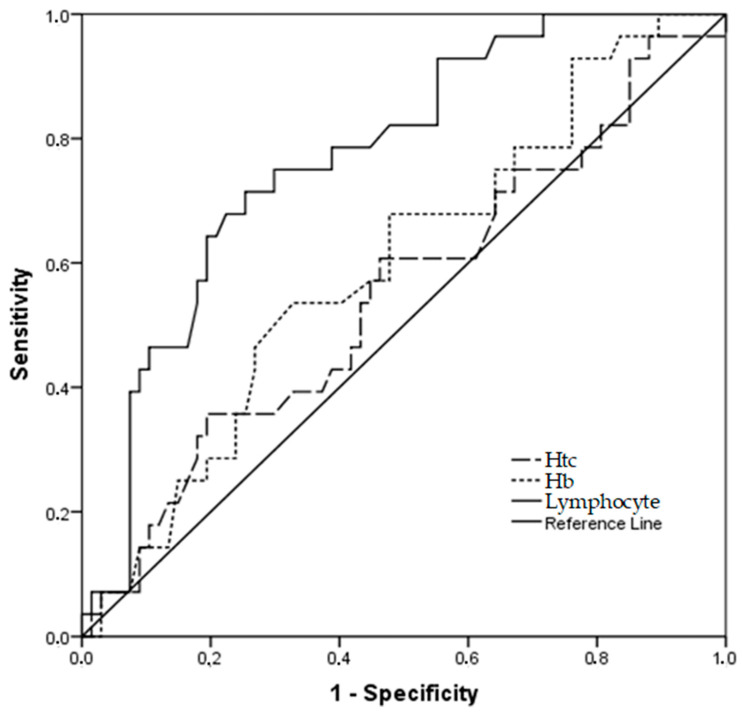
ROC curves in estimating sequelae rates related to Htc, Hb, and lymphocyte parameters.

**Table 1 vaccines-11-01321-t001:** Descriptive statistics of the general characteristics of the cases.

Parameters	n/M (Q1–Q3) ^1^
Female	43
Male	57
Age (years)	56 (44.5–65)
Sequelae	
Non	74
Mild	13
Severe	13
WBC (×10^3^/µL)	6.4 (4.86–9.7)
Hematocrit (%)	40.94 (36.85–44.05)
Hemoglobin (g/dL)	13.4 (11.4–14.64)
Platelet (×10^3^/µL)	183 (160–253)
Lymphocyte (×10^3^/µL)	1.2 (0.92–1.49)
Neutrophil (×10^3^/µL)	4.39 (3.05–7.49)
NLR ^2^	3.39 (2.31–6.71)
CRP (mg/L)	53.25 (24–105.5)
D-Dimer	392 (226–704)
Ferritin (ng/mL)	296 (156–719)
Procalcitonin (ng/mL)	0.1 (0.05–0.15)
Fibrinogen (mg/dL)	77.5 (74–80)
Severe COVID-19 pneumonia	23
Mild–moderate COVID-19 pneumonia	77
No vaccine	62
Vaccinated	38
Sinovac	
No vaccine	70
Single dose	10
Two and above	20
Biontech	
No vaccine	86
Single dose	5
Two and above	9

^1^ M: median, Q1: percentile 25, Q3: percentile 75; ^2^ NLR: neutrophil to lymphocyte ratio.

**Table 2 vaccines-11-01321-t002:** The relationship between the general and clinical features of the cases and the presence or absence of sequelae.

	Sequelae + (n = 26)	No Sequelae (n = 74)	
	n (%)/M(Q1–Q3) ^1^	n (%)/M(Q1–Q3) ^1^	*p*-Value
Sex			
Female	10 (38,4)	33 (44.5)	0.692 ^4^
Male	16 (61,5)	41 (55.4)	
Age (years)	54 (44–65)	56 (45–66)	0.310 ^3^
WBC (×10^3^/µL)	6.69 (5.2–12.21)	6.1 (4.79–8.87)	0.149 ^3^
Htc (%)	41.25 (37.2–45)	40.25 (36.85–43.75)	0.406 ^3^
Hb (g/dL)	13.85 (11.5–14.8)	13.3 (11.3–14.4)	0.239 ^3^
Plt (×10^3^/µL)	193.5 (165.5–254)	176 (155–243)	0.250 ^3^
Lymphocyte (×10^3^/µL)	1.44 (1.11–1.9)	1.12 (0.76–1.39)	**0.001** ^3^
Neutrophil (×10^3^/µL)	4.72 (3.26–9.8)	4.25 (3.01–7.12)	0.274 ^3^
NLR ^2^	3.10 (1.75–5.96)	3.50 (2.41–7)	0.099 ^3^
CRP (mg/L)	56.5 (24.5–98.5)	53.25 (23.5–107)	0.726 ^3^
D-dimer	497.5 (219–665)	392 (239–704)	0.802 ^3^
Ferritin (ng/mL)	403 (232–733)	295 (147.5–657)	0.438 ^3^
Procalcitonin (ng/mL)	0.11 (0.06–0.15)	0.1 (0.05–0.16)	0.672 ^3^
Fibrinogen (mg/dL)	79 (75–82)	77 (73–80)	0.185 ^3^
Vaccinated	4 (15.3)	40 (54)	**<0.001** ^4^
No vaccine	22 (84.6)	34 (45.9)	
Sinovac			
No vaccine	25 (96.2)	45 (60.8)	**0.001** ^4^
Single dose	0 (0)	10 (13.5)	
Two and above	1 (3.8)	19 (25.7)	
Biontech			
No vaccine	23 (88.5)	63 (85.1)	0.321 ^4^
Single dose	0 (0)	5 (6.8)	
Two and above	3 (11.5)	6 (8.1)	

^1^ M: median, Q1: percentile 25, Q3: percentile 75; ^2^ NLR: neutrophil to lymphocyte ratio; ^3^
*p* value was obtained from Mann–Whitney U or Student’s *t*-test; ^4^ *p* value was obtained from chi-square or Fisher’s exact test.

**Table 3 vaccines-11-01321-t003:** Relationship between sequelae (absent–mild/moderate and severe) levels and vaccination status.

	Sequelae			
	Non (n = 74)	Mild/Moderate (n = 13)	Severe (n = 13)	
	n (%)	n (%)	n (%)	*p*-Value
Vaccinated	35 (47.3)	3 (23)	1 (7.7)	**0.005**
Unvaccinated	39 (52.7)	10 (77)	12 (92.3)	
Sinovac				
No vaccine	45 (60.8)	12 (92.3)	13 (100)	0.019
Single dose	10 (13.5)	0 (0)	0 (0)	
Two and above	19 (25.7)	1 (7.7)	0 (0)	
Biontech				
No vaccine	63 (85.1)	11 (84.6)	12 (92.3)	0.880
Single dose	5 (6.8)	0 (0)	0 (0)	
Two and above	6 (8.1)	2 (15.4)	1 (7.7)	

**Table 4 vaccines-11-01321-t004:** Investigation of the relationship between sequelae status, blood parameters, and fev/fvc values with ROC curves.

Test Result Variable(s)	AUC ^1^ (95% CI ^2^)	Std. Error	*p*-Value	Sensitivity	Specificity
Htc > 44.15	0.76 (0.61–0.91)	0.076	**0.005**	0.457	0.806
Hb >13.95	0.79 (0.66–0.93)	0.069	**0.001**	0.536	0.672
Lymphocyte > 1.36	0.73 (0.57–0.89)	0.081	**0.014**	0.714	0.746

^1^ AUC: the Area Under the Curve; ^2^ CI: confidence interval.

## Data Availability

The data supporting the findings of this study are available from the corresponding author (S.A.) upon request.

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
