# Peer review of "The Effects of Vaccines on the Sequelae Rates of Recurrent Infections and the Severity of Pulmonary COVID-19 Infection by Imaging"

_vaccines, 2023, doi:10.3390/vaccines11081321_

Round 1

Reviewer 1 Report

The authors used the follow-up CT images after initial hospitalization due to pulmonary COVID infection to contrast the sequelae outcomes between vaccinated and unvaccinated cohorts. The primary finding is that the vaccination status is associated with lower risks and severity of sequelae. While the finding may contribute to the continuous prevention and management of COVID, several aspects in the background, design and statistical analysis need to be clarified or modified.

1.       The study highlighted the use of sequelae from imaging data as the outcome. The sequelae outcome, however, is a short term measurement within 6 months. To justify that this is better than clinical and laboratory outcomes, authors should demonstrate its role in long term outcomes, e.g. mortality or disability.

2.       Authors should offer details on the recruitment process and assess the risk of sampling bias. Since the study is a retrospective investigation of patients hospitalized between April 2020 and August 2021 at the end of this window, deceased patients at the start of the study cannot be recruited due to inability to obtain their written consent. No doubt there is no death reported over the 100 recruited patients. The vital status of the total 900 patients may serve as a reference, or authors should refer to mortality reported over comparable cohorts in existing literatures.

3.       The inclusion criterion regarding complete CT follow-ups needs to be clarified. Is this a standard protocol for all patients? Authors should demonstrate if there is a risk of sampling bias. For example, the conclusion can be overturned if many healthy unvaccinated patients were excluded from the study because they did not come back for those CTs.

4.       Characteristics derived from imaging are sometimes ambiguous.  The authors should consider justify the consistency of their outcomes by reporting agreements between two separate radiologists as in many other studies.

5.       The study has conducted a lot of statistical testing. Authors should filter potentially spurious findings by adjusting for multiple testing.

6.       To predict sequelae, authors should consider a model aggregating all predictive markers. I am not sure how useful the three separate associations can be in practice.

7.       COVID was a rapidly evolving disease during the study period. Associations may be distorted if variables shifted across time. Authors should report temporal trajectories and/or adjust for temporal trends in their analysis as sensitivity check.

Author Response

Dear Editor,

Thank you for giving us the opportunity to submit a revised draft of the manuscript. We appreciate the time and effort that you and the reviewers dedicated to providing feedback on our manuscript and are grateful for the insightful comments on and valuable improvements to our paper. We have incorporated most of the suggestions made by the reviewers. Those changes are highlighted with track changes function through the manuscript. Please see below for a point-by-point response to the reviewers’ comments and concerns.

Reviewer 1 report:

Comments and Suggestions for Authors

The authors used the follow-up CT images after initial hospitalization due to pulmonary COVID infection to contrast the sequelae outcomes between vaccinated and unvaccinated cohorts. The primary finding is that the vaccination status is associated with lower risks and severity of sequelae. While the finding may contribute to the continuous prevention and management of COVID, several aspects in the background, design and statistical analysis need to be clarified or modified.

  1. The study highlighted the use of sequelae from imaging data as the outcome. The sequelae outcome, however, is a short term measurement within 6 months. To justify that this is better than clinical and laboratory outcomes, authors should demonstrate its role in long term outcomes, e.g. mortality or disability.
  2. Authors should offer details on the recruitment process and assess the risk of sampling bias. Since the study is a retrospective investigation of patients hospitalized between April 2020 and August 2021 at the end of this window, deceased patients at the start of the study cannot be recruited due to inability to obtain their written consent. No doubt there is no death reported over the 100 recruited patients. The vital status of the total 900 patients may serve as a reference, or authors should refer to mortality reported over comparable cohorts in existing literatures.
  3. The inclusion criterion regarding complete CT follow-ups needs to be clarified. Is this a standard protocol for all patients? Authors should demonstrate if there is a risk of sampling bias. For example, the conclusion can be overturned if many healthy unvaccinated patients were excluded from the study because they did not come back for those CTs.
  4. Characteristics derived from imaging are sometimes ambiguous.  The authors should consider justify the consistency of their outcomes by reporting agreements between two separate radiologists as in many other studies.
  5. The study has conducted a lot of statistical testing. Authors should filter potentially spurious findings by adjusting for multiple testing.
  6. To predict sequelae, authors should consider a model aggregating all predictive markers. I am not sure how useful the three separate associations can be in practice.
  7. COVID was a rapidly evolving disease during the study period. Associations may be distorted if variables shifted across time. Authors should report temporal trajectories and/or adjust for temporal trends in their analysis as sensitivity check.

Response:

  1. We thank the reviewer for the valuable comment. We stated the situation as a limitation: “It is an undeniable fact that 6-month short-term imaging results will not be sufficient for evaluation in the follow-up of pulmonary sequelae after pulmonary COVID-19 infection. However, we did not have sufficient data to evaluate the long-term results, as most of the patients stopped coming to the hospital control after the end of the pandemic. In addition, there is a systematic and sampling bias because patients with severe and critical COVID 19 pneumonia, most of whom resulted in death and disability, could not be included in our study since 6-month follow-up images were not available.”

  1. The following sentence is added to the materials and methods section for the details of the recruitment process.

“Among these 900 patients, 100 patients with 1st, 3rd and 6th month follow-up thoracic CT examinations within the specified date ranges were included in our study.”

Since patients who could undergo follow-up thorax CT scans for pulmonary sequelae were included in our study, none of the cases resulting in death due to COVID 19 pneumonia were included in our study for which follow-up thorax CT images could not be obtained.

The following sentence has been added to the discussion section to provide information on the vital status of the total of 900 patients.

“We found a mortality rate of 1.1% in 900 patients we screened. According to the WHO dashboard, although this rate is above the rate stated for Turkey, it is similar to the global mortality rates.”

  1. This is stated as a limitation. “Inclusion criterion is not a standard protocol. It is a sample taken from patients who were admitted to our hospital. These findings may not be generalizable because they only comprise patients who had follow-up thoracic CT scans.”

  1. We apologize for the mistake in the materials and methods section regarding the evaluation of images. The following sentences have been added.

“ All the chest CT images were evaluated by 2 board-certified radiologists in consensus. Therefore, there is no inter-observer disaggreement.”

  1. Which test is used for each parameter is marked on the tables and added to the explanations.

The parameters for which ROC analysis and sensitivity-specificity are used, added to the statistical aproach section.

  1. “In the time period when we planned our work, there was no study in the literature describing sequelae findings in imaging after COVID-19 pneumonia. In our study, sequelae were described as band like atelectasis, traction bronchiectasis and focal ground glass areas similar to other patterns of fibrosis mostly seen in peripheral/ sub-pleural and basal parts of the lungs. These findings were similar to the most common sequelae of pulmonary parenchymal changes that Watanabe et al. described in their study. In this regard, our study complies with a standardization in the literature.” This sentence is added to the discussion section

  1. The time of obtaining blood parameters was expressed in detail in the materials and methods section and we stated the situation as a limitation: “…the examined blood parameters were acquired within three days of the first diagnosis, we cannot offer any results about the temporal change of these parameters and/or the relationship between this change and the sequelae formation.”

Reviewer 2 Report

1. The conclusion is not reliable with the results of study.

2. The reason for conducting this work is not well presented in Introduction

3. There are several concerns about methodology of this paper

Firstly, where was conducted this study (I request it by regarding the affiliations of authors)

What are criteria of COVID-19 classification and sequelae classification

The authors conducted this study among only patient at hospital admission, hence there is a systematic bias that could effect the conclusion. Because the mild or asymptomatic patients might be treated at home.

What are differences of indication for vaccination with Sinovac and Biotech?

 There were 74 patient with no sequelae and 26 with sequelae, but in table 2, there were 70 and 30, respectively

When investigating the relationship between sequelae and blood testing, why the authors selected only Hct, Hb and Lymphocytes? Why they did not consider D-dimers, CRP and others important markers?

Author Response

Dear Editor,

Thank you for giving us the opportunity to submit a revised draft of the manuscript. We appreciate the time and effort that you and the reviewers dedicated to providing feedback on our manuscript and are grateful for the insightful comments on and valuable improvements to our paper. We have incorporated most of the suggestions made by the reviewers. Those changes are highlighted with track changes function through the manuscript. Please see below for a point-by-point response to the reviewers’ comments and concerns.

Reviewer 2 report:

Comments and Suggestions for Authors

  1. The conclusion is not reliable with the results of study.
  2. The reason for conducting this work is not well presented in Introduction
  3. There are several concerns about methodology of this paper

Firstly, where was conducted this study (I request it by regarding the affiliations of authors)

What are criteria of COVID-19 classification and sequelae classification

The authors conducted this study among only patient at hospital admission, hence there is a systematic bias that could effect the conclusion. Because the mild or asymptomatic patients might be treated at home.

What are differences of indication for vaccination with Sinovac and Biotech?

There were 74 patient with no sequelae and 26 with sequelae, but in table 2, there were 70 and 30, respectively

When investigating the relationship between sequelae and blood testing, why the authors selected only Hct, Hb and Lymphocytes? Why they did not consider D-dimers, CRP and others important markers?

Response:

  1. By editing the conclusion part, we attempted to write a more reliable conclusion supported by the findings of our study.

“In conclusion, our study showed that a high frequency of residual CT abnormalities and these sequelae findings are seen more frequently in unvaccinated cases months after COVID-19. Vaccination not only provides protection against infection, but also reduces the rate of sequelae following possible infection. Further investigations with longer follow‐up periods will help understand the underlying mechanism of COVID‐19 sequelae and manage patients with these conditions.”

  1. We have revised the introduction to include more information to support our reasons for conducting this study. The following sentences have been added to this context.

“Because previous SARS and MERS outbreaks indicated that symptoms and imaging abnormalities persist over time, it has been proposed to monitor patients following acute SARS-CoV-2 pneumonia. In 2020, the British Thoracic Society (BTS) published a document for post-COVID-19 care that distinguished between patients with severe pneumonia and those with mild-moderate pneumonia. The goal of this document was to standardize radiological follow-up and then reduce the pressures on respiratory services following the initial COVID-19 outbreak”

“Previous studies investigated lung sequelae of COVID‐19, demonstrated that more than half of the recovered patients showed thorax CT abnormalities months after infection. Groundglass opacities (GGOs), parenchymal bands/fibrous stripes, and reticulation were particularly common.”

  1. Please see below for our responses to the reviewer's various concerns about the methodology of this article.

***This study was carried out in a university hospital, which is a tertiary referral center in our country.

*** “Based on the Guideline for the Diagnosis and Treatment Plan of COVID-19 Infection by the National Health Commission, pulmonary COVID-19 infection was classified into three types: mild to moderate type common with fever, respiratory symptoms and imaging presentations of pneumonia; severe type with any of the following: respiratory distress with RR> 30 times/minutes, oxygen saturation at rest <93%, or PaO2/FiO2<300 mmHg (1 mmHg=0.133 kPa); critically severe type with any of the following: respiratory failure needing mechanical ventilation, shock, or combination with other organ failure needing ICU intensive care.” This sentence is added to the materials ans methods section.

“In the time period when we planned our work, there was no study in the literature describing sequelae findings in imaging after COVID-19 pneumonia. In our study, sequelae were described as band like atelectasis, traction bronchiectasis and focal ground glass areas similar to other patterns of fibrosis mostly seen in peripheral/ sub-pleural and basal parts of the lungs. These findings were similar to the most common sequelae of pulmonary parenchymal changes that Watanabe et al. described in their study. In this regard, our study complies with a standardization in the literature.” This sentence is added to the discussion section

*** We thank the reviewer for the valuable comment. We stated the situation as a limitation: “We generally conducted this study on patients who were admitted to the hospital with thoracic CT imaging, although rarely, patients who received home treatment came to the hospital for thoracic CT control and were included in our study, but almost all mild or asymptomatic COVID-19 patients were treated at home.” Therefore, there is a systematic and sampling bias in our study that could be affect the result.

*** Sinovac and Biontech vaccines are made according to the preferences of the people in our country. There is no indication determined by our Ministry of Health in this regard. The first vaccine supplied to our country during the pandemic period was Sinovac, about 6 months before Biontech. For this reason, there are people who take a booster dose with Biontech after vaccination with Sinovac.

*** We apologize for the mistake in the Table 2. The number and percentages of no sequelae and sequelae patients in Table 2 have been corrected as shown below.

Vaccinated

3 (11.5)

35 (47.3)

<0.001

No Vaccine

23 (88.5)

39 (52.7)

Sinovac

No Vaccine

25 (96.2)

45 (60.8)

0.001

Single dose

0 (0)

10 (13.5)

Two and above

1 (3.8)

19 (25.7)

Biontech

No Vaccine

23 (88.5)

63 (85.1)

0.321

Single dose

0 (0)

5 (6.8)

Two and above

3 (11.5)

6 (8.1)

*** Our goal in investigating the association between sequelae and blood tests using only Hct, Hb, and Lymphocyte counts is to collect enough data to reach statistically significant results in these three parameters. Since we could not obtain D-dimer, CRP and other important markers in every patient, we could not reach data that could obtain meaningful results.

Round 2

Reviewer 1 Report

The authors clarified their study in the revision. 

Author Response

Dear Reviewer,

We appreciate the time and effort that you dedicated to providing feedback on our manuscript and are grateful for the insightful comments and valuable improvements to our paper. We have made the relevant corrections according to your suggestions. Please see below for a point-by-point response to the comments and suggestions.

Thank you for the encouraging comments and all of your valuable efforts.

Reviewer 2 Report

As my first round of review, the paper has several concerns.

The authors did not respond satisfactorily to my comments, especially:

1. The revised conclusion: "In conclusion, our study showed that a high frequency of residual CT abnormalities and these sequelae findings are seen more frequently in unvaccinated cases months after COVID-19. Vaccination not only provides protection against infection, but also reduces the rate of sequelae following possible infection. Further investigations with longer follow‐up periods will help understand the underlying mechanism of COVID‐19 sequelae and manage patients with these conditions."

However, they did not evaluate the association between vaccination and infection. Furthermore, they can not conclude that vaccination reduces the rate of sequelae following possible infection based only univariate analysis. Moreover, there are several important variables such as D-Dimer, Fibrinogen, Platelets were not investigated.

2. where was conducted this study? The author's response is that this study was carried out in a university hospital, which is a tertiary referral center in our country. But I see there are authors from Australia and Turkey. So, please provide the name of the hospital and country. 

3. I still have a major concern, regarding the Table 2. I doubt the accuracy of the data.

In the Results, there were 26 patients with sequelae and 74 without sequelae. But in the Table 2 "Sequelae + (n=30)"; "No sequelae (n=70)". Indeed, the total of sex in these two group were 30 and 70.

There were in total 3 vaccinated patients with sequelae, but 1 patient was vaccinated with Sinovac and 3 others with Biontech (N = 1+3=4). Similarly, there were 35 vaccinated patients without sequelae, but in total, 29 were vaccinated with Sinovac and 11 with Biontech (N = 29+11=40)

Author Response

Dear Reviewer,

We appreciate the time and effort that you dedicated to providing feedback on our manuscript and are grateful for the insightful comments and valuable improvements to our paper. We have made the relevant corrections according to your suggestions. Please see below for a point-by-point response to the comments and suggestions.

The authors did not respond satisfactorily to my comments, especially:

  1. The revised conclusion: "In conclusion, our study showed that a high frequency of residual CT abnormalities and these sequelae findings are seen more frequently in unvaccinated cases months after COVID-19. Vaccination not only provides protection against infection, but also reduces the rate of sequelae following possible infection. Further investigations with longer follow‐up periods will help understand the underlying mechanism of COVID‐19 sequelae and manage patients with these conditions."

However, they did not evaluate the association between vaccination and infection. Furthermore, they can not conclude that vaccination reduces the rate of sequelae following possible infection based only univariate analysis. Moreover, there are several important variables such as D-Dimer, Fibrinogen, Platelets were not investigated.

Response: The conclusion is rewritten according to the recommendations: “In conclusion, our study showed that a high frequency of residual CT abnormalities and these sequelae findings are seen more frequently in unvaccinated cases months after COVID-19. Even though we cannot evaluate the combined effect of multiple laboratory parameters on sequelae formation, with the presented data one can infer that vaccination can reduce the rate of sequelae following possible infection. Further investigations with longer follow‐up periods will help understand the underlying mechanism of COVID‐19 sequelae and manage patients with these conditions”

  1. where was conducted this study? The author's response is that this study was carried out in a university hospital, which is a tertiary referral center in our country. But I see there are authors from Australia and Turkey. So, please provide the name of the hospital and country. 

Response: The relevant sentrence is revised as recommended: “This study was carried out in a university hospital, which is a tertiary referral center in our country (Baskent University Hospital, Ankara, TURKEY).”

  1. I still have a major concern, regarding the Table 2. I doubt the accuracy of the data.

In the Results, there were 26 patients with sequelae and 74 without sequelae. But in the Table 2 "Sequelae + (n=30)"; "No sequelae (n=70)". Indeed, the total of sex in these two group were 30 and 70.

There were in total 3 vaccinated patients with sequelae, but 1 patient was vaccinated with Sinovac and 3 others with Biontech (N = 1+3=4). Similarly, there were 35 vaccinated patients without sequelae, but in total, 29 were vaccinated with Sinovac and 11 with Biontech (N = 29+11=40)

Response: Many thanks for the attentioan and the warning. The mistake was happened during the data transfer from SPSS datasheets to the article. We have corrected the number inconsistencies.
